# OpenReview forum: "SafeMind: Benchmarking and Mitigating Safety Risks in Embodied LLM Agents"
_ICLR.cc/2026/Conference — ICLR 2026 Conference Withdrawn Submission_

### Official Review · Reviewer_NXfs · 2025-10-15

**Soundness:** 3
**Presentation:** 3
**Contribution:** 3
**Rating:** 6
**Confidence:** 3

**Summary:**

This paper presents SafeMind, a comprehensive framework for analyzing and mitigating safety risks in embodied large language model (LLM) agents.
It formalizes a four-stage reasoning pipeline—Task Understanding, Environment Perception, High-Level Plan Generation, and Low-Level Action Generation—and defines three orthogonal constraint types (Factual, Causal, Temporal) to categorize safety violations.

Based on this taxonomy, the authors build SafeMindBench, a multimodal benchmark comprising 5,558 instruction–image pairs spanning four categories: Instr-Risk, Env-Risk, Order-Fix, and Req-Align. They also propose SafeMindAgent, a modular Planner–Executor architecture integrating three cascaded safety modules (Task-Safe, Plan-Safe, Action-Safe) with a Safety Constraint Knowledge Base (SCKB).

Experiments on leading MLLMs (GPT-4o, Claude-Sonnet-3.7, Gemini-2.5-Flash, etc.) and agent architectures (ReAct, Plan-and-Act, MLDT) show that SafeMindAgent improves safety rate by over 24% on average without sacrificing task success, outperforming prior agents on both SafeMindBench and SafeAgentBench

**Strengths:**

Well-structured conceptual framework – The four-stage pipeline and tri-constraint taxonomy provide a clear and systematic safety model applicable to diverse embodied reasoning agents.

Benchmark contribution – SafeMindBench fills a major gap in multimodal embodied safety testing by combining textual and visual risk scenarios under real-world categories (sabotage, harm, privacy, illegal behavior).

**Weaknesses:**

(1) Limited coverage of backdoor and adversarial threat literature.
While the paper focuses on inherent safety risks, it should cite complementary research showing malicious manipulation of embodied agents, e.g.:

TrojanRobot: Physical-World Backdoor Attacks Against VLM-Based Robotic Manipulation (arXiv:2411.11683) — demonstrates real-world backdoors using visual triggers.



(2) Unclear Formulation of Evaluation Metrics. It is suggested to formuate the evaluation metrics to better reflect the expressions in Section 4.3.

 (3) Constraint interpretation dependency.
The evaluation still depends on GPT-4 as an LLM-judge, which introduces uncertainty and potential bias in safety scoring.

**Questions:**

NA

---

### Official Review · Reviewer_zwUK · 2025-10-29

**Soundness:** 2
**Presentation:** 2
**Contribution:** 1
**Rating:** 2
**Confidence:** 5

**Summary:**

This paper introduces "SafeMind", a framework to benchmark and mitigate safety risks in embodied LLM agents. The authors formalize a four-stage risk model and introduce SafeMindBench, a 5,558-sample multimodal benchmark designed to evaluate safety vulnerabilities at each stage. To address these, they propose SafeMindAgent, a Planner-Executor architecture with three cascaded safety modules and an external knowledge base (SCKB) that uses multi-stage verification. Experiments show SafeMindAgent improves safety rates over baselines while maintaining task performance.

**Strengths:**

-The paper tackles the safety of embodied LLM agents, which is a critical and highly relevant research problem. As these agents are poised for real-world deployment, ensuring their safety is a paramount and non-trivial challenge.


- The paper is supported by a comprehensive set of experiments, including evaluations of both standalone MLLMs and various agent architectures.

**Weaknesses:**

## **1. Limited Novelty and Contribution Over Prior Work**

The paper's claimed contributions do not feel sufficiently novel.

- **The proposed four-stage risk model is not a new contribution:** The authors present the decomposition of the agent's reasoning process into four stages (Task Understanding, Environment Perception, High-Level Plan Generation, Low-Level Action Generation) as a key contribution. However, this is a widely accepted and almost standard conceptual framework in the embodied reasoning literature, not a novel formalization. Many prior works implicitly or explicitly adopt this pipeline.

- **The SafeMindBench  offers limited methodological advancement:** The benchmark itself does not offer a fundamental methodological leap over existing datasets. The strategy of using LLMs to generate text-instructions and leveraging text-to-image models to create corresponding static scenes is a similar paradigm to that used by EARBench (which the authors cite). The paper fails to clearly articulate what makes SafeMindBench qualitatively different from EARBench or SafeAgentBench beyond simply having more samples or different task labels. The "stage-isolation" seems to be a post-hoc labeling of tasks rather than a truly novel benchmark design.

## **2. Unrealistic and Impractical Agent Architecture**

- The SafeMindAgent is built on a Planner-Executor architecture where both components are implemented as LLMs. This is a profound misunderstanding of the "Executor" role in embodied AI.
In robotics, an executor's function is to ground a high-level symbolic plan (e.g., "pick up the red apple") into a sequence of low-level, continuous control signals or actions (e.g., joint velocities, end-effector poses, gripper commands). This grounding task is a major research challenge in itself, typically handled by models trained on large-scale robotic data, such as **Vision-Language-Action (VLA) models or Diffusion Policies**. A pre-trained LLM, without specialized training, is fundamentally incapable of generating these control signals.

- The paper's Executor (as detailed in Appendix C.2) **appears to be another LLM call** that just outputs a textual list of action descriptions. This reduces the entire framework to a **text-in, text-out exercise that is completely disconnected from physical reality**. This "armchair agent" design makes it impossible to judge if the agent's "safety" (which is merely safer text generation) has any correlation with the practical safety of a physical robot.

## **3. Static, Open-Loop Evaluation Bypasses Core Embodied Challenges**

- The paper's entire evaluation methodology is **static and open-loop**, which is insufficient for an embodied agent.
The agent is given a single static image, generates a complete plan/action sequence, and this static sequence is then evaluated. This is not how embodied agents operate.

- Embodied AI is inherently **dynamic and closed-loop**. An agent must execute one action, observe the new state of the world (which has changed as a consequence of its action), and then re-plan or select its next action based on this new feedback.
The paper's static evaluation completely ignores reacting to the consequences of one's own actions and to unexpected environmental changes. A plan deemed "safe" from a single static image may become instantly unsafe after the first step. This static methodology is a significant shortcoming.

## **4. Lack of Simulator or Real-World (Sim2Real) Validation**
- The paper's evaluation is limited to the text and image level.

- For a problem as dynamic as embodied safety, the benchmark scenarios should have been constructed within a standard simulator (e.g., AI2-THOR, VirtulHome). This would have allowed for proper, closed-loop evaluation where the agent's actions have tangible consequences on the environment. Furthermore, a paper claiming to mitigate safety risks should, at a minimum, demonstrate Sim2Real transfer or, ideally, include experiments on a real robotic platform. Without this, the claims of improved "safety" remain purely theoretical and unverified in the context of real-world robotics.

## **5. Unanalyzed Inefficiency and System Brittle-ness**

- **The SafeMindAgent architecture introduces significant computational overhead.**
It proposes a cascade of safety modules ($M_T$, $M_P$, $M_A$), each potentially being a separate LLM call. Embodied agents must operate with low latency. The paper provides no analysis of the latency or computational cost added by this multi-stage verification, making its practicality highly questionable.

## **6.Irrelevant Task Category in the Safety Benchmark**

- The fourth task Req-Align (Requirement-Alignment), appears to be misplaced in a safety benchmark.
The example given ("Ensure that the oil is poured... no more than two steps after switching on the stove") is a test of procedural adherence, complex instruction following, and temporal planning accuracy.

- This is fundamentally a test of the agent's capability and fidelity to a task, not its ability to mitigate harm. While failing this task might result in a poorly cooked meal, it does not fit the same risk profile as electrocution, harming a pet, or starting a fire. Including this task dilutes the benchmark's focus on safety.

**Questions:**

Refer to Weaknesses

---

### Official Review · Reviewer_sLo3 · 2025-10-29

**Soundness:** 2
**Presentation:** 3
**Contribution:** 2
**Rating:** 2
**Confidence:** 3

**Summary:**

This paper proposes a benchmark called SafeMindBench, evaluating the safety awareness of embodied agents when interacting with physical environments. The benchmark design risk models cover tasks including task understanding, environment perception, high-level plan generation, and low-level plan generation. The paper further proposes SafeMindAgent to improve the safety awareness by reflecting the whole decision-making process. Experiments on 5,558 multimodal tasks show that existing MLLMs and agent frameworks exhibit poor safety awareness (safety rates below 50%), while SafeMindAgent improves safety by 25.8% over the best baseline without sacrificing success rate.

**Strengths:**

- **Interesting topics**: The paper explores an interesting topic of the safety awareness of embodied agents, which is critical for the real-world deployment of embodied AI.
- **Well-defined risk model**: The main difference between the paper and the previous benchmark is the clean definition of the risk model throughout the decision-making process, from task understanding to the final execution. These definitions can help pinpoint which stage causes the final unsafe behaviors of embodied agents.
- **Promising results**: The paper shows that by reflecting on every step of the agent's thinking process, we can improve the safety awareness measured by the proposed benchmarks.

**Weaknesses:**

## Major


- **Lack of closed-loop setups**:
  - **Evaluation**: While the benchmark and agent design are conceptually sound and supported by systematic experiments across multiple baselines, the evaluation relies solely on textual generation without closed-loop validation, which limits real-world applicability.
  - **Ground Truth Generation**: Additionally, the ground truth action is also generated by LLMs. Even though the skill set is used to check the data quality (Sec 4.4), it only checks the availability of the action, not the executability given specific environmental constraints. Thus, the ground truth quality is also not guaranteed to be at least executable.
  - **Suggestions**: Since the benchmark is for an LLM agent, it’s suggested that the author go beyond textual-level evaluation and perform experiments on simulators (e.g., VirtulHome) to justify the benchmark's soundness (transferable conclusions) and the SafeMindAgent's effectiveness.
- **Lack of detailed data quality justification**:
  - **Justification for human verifier setups**: The data is collected from LLMs, followed by humans verifying the data quality, which is good to ensure the data quality. However, no details are provided on the human verification process. For example, how many human laborers are used? Are there any detailed instructions on human of how to determine the data quality to judge what data should be discarded? Without knowing the detailed benchmark construction process. It’s unknown if the benchmark quality is high enough to derive meaningful results. Additionally, no justification is provided for the data diversity.
  - **Justification for data diversity**: The paper doesn't discuss the data diversity. Therefore, it's unknown if the 5,558 pieces of data are highly diverse or largely overlapping in terms of the tasks.
  - **Suggestions**: The author is suggested to provide detailed human verification setups and justification of the data diversity. For example, [r1] uses ROUGE-L similarity to filter out repetitive data.

## Minor

- **Repetitive figures**: Figure 4a and Table 3 present overlapping data; combining them could improve conciseness.
- **Lack of data in the supplementary materials**: Since it’s a benchmark paper, it’s beneficial to upload the data as one of the supplementary materials. Without this, it’s hard to judge the data quality.

---
**Reference**

[r1] Wang, Yizhong, et al. "Self-instruct: Aligning language models with self-generated instructions." arXiv preprint arXiv:2212.10560 (2022).

**Questions:**

Please see the weakness part.

---

### Official Review · Reviewer_bREE · 2025-10-31

**Soundness:** 2
**Presentation:** 3
**Contribution:** 2
**Rating:** 4
**Confidence:** 4

**Summary:**

This paper proposes a framework for benchmarking and mitigating safety risks in embodied LLM agents that interact with the physical world. The authors formalize safety vulnerabilities across a four-stage reasoning pipeline and define three orthogonal constraint types. They demonstrate that  LLMs like GPT-4o and existing agent architectures show low average safety rates below 40%, and their method improves the average safety rate by 24.5% while maintaining comparable task completion rates (93.8%).

**Strengths:**

This paper addresses an emerging and important area of research: the safety of embodied LLM agents. As our community and society pay close attention to this area, I am happy to see a paper submission in this area. Their evaluation shows high improvements from the baseline and existing methods. They demonstrate that  LLMs like GPT-4o and existing agent architectures show low average safety rates below 40%, and their method improves the average safety rate by 24.5% while maintaining comparable task completion rates (93.8%). They evaluate their benchmark on 5 LLM agents.

**Weaknesses:**

I have the following major concerns about this paper:

### Insufficient justification of why their new benchmark is necessary

I am still concerned about the necessity for creating yet another safety benchmark, given the existence of multiple recent benchmarks as listed in Table 1. While this paper claims that their benchmark can cover more risk categories, stage isolation, and higher realism, this paper does not answer how these aspects are significant to improve the safety of embodied LLM agents. For example, the paper claims SafeAgentBench and IS-Bench have "low realism" due to objects floating in mid-air in simulators; I am not sure their DALL-E 3-based image generation can address it. The authors argue that existing benchmarks suffer from limitations such as text-only modalities, low realism in simulators, single-stage hazards, and la ack of process-oriented evaluation, yet these criticisms are not evidenced by concrete experimental analysis.

### Potential data leakage in SCKB and concerns on its generality

I am concerned about the potential leakage in SCKB because the SCKB itself is generated by the 300 tasks sampled from their dataset. While the authors claim "the evaluation set is constructed to exclude all task instances used during constraint extraction to avoid information leakage", there might be a potential high correlation between the data used in the evaluation and used for the SCKB. To address the concern, this paper should conduct the evaluation on multiple benchmarks, such as the ones in Table 1. This paper may also conduct cross-benchmark evaluation, i.e., they may construct the SCKB on their dataset and evaluate on the other datasets.

### Lack of comparison with safety-specific baselines

I appreciate their evaluation of the 5 existing agents. However, I am concerned that their evaluation does not cover recent safety-focused methods explicitly mentioned in the related work section, such as Pinpoint, Safe-BeAl, and Concept Enhancement Engineering. The SafeAgentBench paper also proposes ThinkSafe. Without such comparisons, I cannot be fully convinced whether their improvements are rooted in their original designs or existing designs known in this area.

**Questions:**

- Are there any particular reasons why this paper does not include a comparison with safety-specific baselines? Are there any obstacles to applying these baselines to their cases?
- Is there any significant case where their method can prevent critical safety issues?

---

### Note · Authors · 2025-12-01

I have read and agree with the venue's withdrawal policy on behalf of myself and my co-authors.